

# Validation of the Malay version of the Diabetes Health Literacy Scale among Malaysian adults with type 2 diabetes mellitus

Pei Ling Ng[1], Wan Nor Arifin[1], Wan Mohd Izani Wan Mohamed[2], Rosnani Zakaria[3] and Yee Cheng Kueh[1]

[1] Biostatistics and Research Methodology Unit, School of Medical Sciences, Universiti Sains Malaysia, Kubang Kerian, Kelantan, Malaysia

[2] Department of Internal Medicine, School of Medical Sciences, Universiti Sains Malaysia, Kubang Kerian, Kelantan, Malaysia

[3] Department of Family Medicine, School of Medical Sciences, Universiti Sains Malaysia, Kubang Kerian, Kelantan, Malaysia

Corresponding author
Yee Cheng Kueh, yckueh@usm.my

## ABSTRACT

**Background.** The aims of this study were to translate and adapt the Diabetes Health Literacy Scale (DHLS) to Malay, and to determine the validity and reliability of the Malay version of the DHLS (DHLS-M) among Malaysian adults with type 2 diabetes mellitus (T2DM).

**Methods.** The DHLS was translated and culturally adapted to Malay, followed by a cross-sectional study which was conducted using a self-administered questionnaire among the adults with T2DM in Hospital Universiti Sains Malaysia (USM). The participants were recruited by convenience sampling. Confirmatory factor analysis (CFA) and correlation analysis were performed.

**Results.** A total of 250 adults with T2DM participated in this study. The median age of the participants was 63.0 years old (IQR = 12.3) and most of the participants were male (51.2%). The final measurement model of DHLS-M with removal of one problematic item, fit the data well based on several fit indices: Relative chi-square ($\chi^2/df$) = 3.858, comparative fit index (CFI) = 0.981, Tucker–Lewis index (TLI) = 0.976. The composite reliability of the three subscales based on Raykov's rho were 0.962, 0.836 and 0.828 respectively. The subscales of DHLS-M were significantly correlated with the Malay version of the Michigan Diabetes Knowledge Test (MDKT) ($r$ = 0.26 to 0.31) and the Malay version of the short form Health Literacy Survey Questionnaire (HLS-SF12) ($r$ = 0.43 to 0.66).

**Conclusion.** The DHLS-M, which consisted of three subscales and 13 items, is a valid and reliable instrument for measuring diabetes health literacy among adults with T2DM in Malaysia. Its validity was further strengthened by the convergent validity with the Malay version of the MDKT and the Malay version of the HLS-SF12.

# INTRODUCTION

Diabetes is a chronic metabolic disorder presented with hyperglycemia. Type 2 diabetes mellitus (T2DM) is the most common types of diabetes in adults (*Goyal, Singhal & Jialal, 2023*). The prevalence of diabetes has increased over the years globally. In 2017, the estimated prevalence of diabetes in the world was 8.8% (*Cho et al., 2018*). The number has increased to 10.5% in 2021 and is expected to increase to 12.2% in 2045 (*Sun et al., 2022*). In Malaysia, the prevalence of diabetes was 17.5% in 2015, increased to 18.3% in 2019 and decreased to 15.6% in 2023 based on the findings from the National Health and Morbidity Survey (*Institute for Public Health, 2015*; *Institute for Public Health, 2020*; *Institute for Public Health, 2024*).

Diabetes requires a lifelong regimen to control the disease and prevent or delay the disease progression and complications. To obtain the optimal control in diabetes, self-management including medication adherence, changes in diet and physical activity, self-monitoring of blood glucose and appointments with healthcare providers, plays a vital role (*Asfaw & Dagne, 2022*; *Gonzalez, Tanenbaum & Commissariat, 2016*; *Sendekie et al., 2022*). It requires an individual to effectively use health information and health services (*Marciano, Camerini & Schulz, 2019*). Studies reported that individuals with higher health literacy have better diabetes knowledge, self-management and glycemic control (*Alsharit & Alhalal, 2022*; *Lee et al., 2021*).

The World Health Organization (WHO) defines health literacy as 'the cognitive and social skills which determine the motivation and ability of individuals to gain access to, understand and use information in ways which promote and maintain good health' (*Nutbeam, 1998*, p. 357). For measuring health literacy, various instruments have been developed worldwide. The instruments commonly used among the people with diabetes can be divided into general health literacy instruments and disease-specific (diabetes) health literacy instruments (*Marciano, Camerini & Schulz, 2019*).

An instrument developed to measure health literacy in one population might not be suitable to use directly in another population due to the variation in the culture, language, healthcare system, or any combination of these. When applying an existing health literacy instrument to a different population that speaks another language, mere translation is inadequate. Cultural adaptation must also be considered to ensure that the translated instrument accurately measures the same health literacy concept and is relevant to the new population (*Beaton et al., 2000*; *Lopatina et al., 2022*; *Zhang et al., 2020*).

In Malaysia, studies reported that more than 60% of the adults with T2DM attended the government healthcare clinics have low health literacy. The general health literacy instruments which have been translated into Malay and validated among Malaysian were used in the studies (*Abdullah et al., 2020*; *Tan & Ismail, 2020*). There is no validated diabetes-specific health literacy instrument in Malay. An instrument measuring health literacy in specific context such as diabetes is important because different contexts may require different abilities and skills (*Pleasant, McKinney & Rikard, 2011*).

The Diabetes Health Literacy Scale (DHLS) is a diabetes-specific health literacy instrument developed by *Lee et al. (2018)* for measuring diabetes health literacy among

adults with T2DM. It is a self-reported instrument, valid and reliable in Korean (*Lee et al., 2018*). The English version of the DHLS has been translated into Persian and validated (*Moshki et al., 2022*). With the high prevalence of diabetes and more than half of the adults with diabetes being uncontrolled, the DHLS is crucial for the Malaysian adults with diabetes (*Institute for Public Health, 2024*). This instrument measures health literacy in the diabetes context. Action can be taken to tailor the educational programs to the health literacy level in order to enhance health literacy, improve self-management and achieve optimal glycemic control among the population.

However, the DHLS is yet to be translated into Malay and validated in the Malaysian population. Therefore, the objectives of this study were to translate and adapt the DHLS to Malay, determine the construct validity and reliability of the Malay version of the DHLS (DHLS-M) among Malaysian adults with T2DM, and examine the convergent validity by correlating the DHLS-M with the Malay version of the Michigan Diabetes Knowledge Test (MDKT) and the Malay version of the short form European Health Literacy Survey Questionnaire (HLS-SF12). The researchers hypothesized that the DHLS-M is valid and reliable in measuring diabetes health literacy among Malaysian adults with T2DM and there is a significant correlation between the score of the Malay version of the MDKT, the Malay version of the HLS-SF12 and the DHLS-M.

# MATERIALS & METHODS

## Study setting and participants

A cross-sectional study was conducted among adults with T2DM receiving treatment at the Family Medicine Clinic and Diabetes Mellitus Specialist Clinic, Hospital Universiti Sains Malaysia (USM) from January to March 2024. The inclusion criteria were Malaysian of age more than 18 years old, diagnosed with T2DM for at least six months and able to read, understand and response to the questionnaire which was in Malay. The sample size of this study was calculated using confirmatory factor analysis (CFA) by root mean square error of approximation (RMSEA) in the Sample Size Calculator (web) by *Arifin (2023)*, which was based on *Kim*'s (*2005*) method. By using expected RMSEA = 0.05, number of items = 14, number of factors = 3, significant level ($\alpha$) = 0.05, power ($1-\beta$) = 80%, expected dropout rate = 30%, the estimated sample size was 278. A total of 278 participants were recruited using convenience sampling in this study.

## Measurement tools

### Socio-demographic and diabetes information

Items related to the participants' socio-demographic characteristics (*e.g.*, age, gender, race, marital status, education level, occupation and monthly income) and diabetes information (treatment regimen, disease duration and HbA1c level) were included in the questionnaire.

### Diabetes Health Literacy Scale

The DHLS was developed to measure diabetes health literacy among adults with diabetes. It consists of 14 items and three subscales. Seven items assess the informational health literacy, four items assess the numerate health literacy, and three items assess the communicative

health literacy. The responses are measured as the level of agreement with a 5-point Likert scale, ranging from zero for 'not really', one for 'slightly', two for 'moderately', three for 'quite a lot' to four for 'very much'. A higher score indicates better health literacy. The Korean version of the DHLS was shown to be valid, with acceptable model fit [relative chi-square ($\chi^2/df$) = 2.41, RMSEA = 0.07, standardized root mean square residual (SRMR) = 0.04, goodness of fit index (GFI) = 0.91 and comparative fit index (CFI) = 0.95] and reliable (Cronbach's alpha for the three subscales were 0.90, 0.80 and 0.85 respectively) (*Lee et al., 2018*). In this study, the English version of DHLS was translated and adapted to Malay as described below.

### Malay version of short form European Health Literacy Survey Questionnaire

The European Health Literacy Survey Questionnaire (HLS-EU-Q47) was developed to measure health literacy among the general population. It consists of 47 items and 12 subscales (*Sørensen et al., 2013*). It was translated into Malay (*Duong et al., 2017*) and a short form, HLS-SF12 was derived from the HLS-EU-Q47. It consists of 12 items and three subscales (healthcare, disease prevention, health promotion). The responses are measured as the level of difficulty with a 4-point Likert scale, ranging from one for 'very difficult', two for 'difficult', three for 'easy' to four for 'very easy'. A higher score indicates better health literacy. Furthermore, the general health literacy index can be calculated using the formula (Index = (mean of items for each person −1) × (50/3)). The index ranged from zero (lowest health literacy) to 50 (highest health literacy). The Malay version of the HLS-SF12 was shown to be valid, with acceptable model fit ($\chi^2/df$ = 2.38, GFI = 0.96, CFI = 0.95 and RMSEA = 0.06) and reliable (Cronbach's alpha for the three subscales were 0.70, 0.73 and 0.71 respectively) (*Duong et al., 2019*).

### Malay version of Michigan Diabetes Knowledge Test

The MDKT was developed to measure diabetes knowledge among adults with diabetes. It consists of 23 items and two subscales (general diabetes knowledge and knowledge of insulin use) (*Fitzgerald et al., 1998*). The general diabetes knowledge subscale, which consists of 14 items, was translated into Malay (*Al-Qazaz et al., 2010*). All the items are multiple-choice questions. One score is given for a correct answer and a zero score for a wrong answer or an unanswered question (*Fitzgerald et al., 1998*). A higher score indicates better diabetes knowledge. The Malay version of the 14-item MDKT was shown to be reliable with Cronbach's alpha 0.70 (*Al-Qazaz et al., 2010*).

## Questionnaire translation and adaptation

In this study, the English version of the DHLS was translated and adapted to Malay based on the procedures in *Wild et al. (2005)* with some modifications. The DHLS was available in the literature and the permission to use the instrument was obtained from the publisher.

The DHLS was forward translated from English into Malay by two independent bilingual translators, who were familiar with the content of the instrument. Both forward translated versions were reconciled into a single forward translation. The single forward translation was then back translated from Malay into English by another two independent bilingual translators. One with medical background while the other was an English language expert.

Next, two experts in questionnaire translation and the researcher reviewed, discussed and harmonized the discrepancies between the single forward translation, two back translations and the original version of the instrument. To ensure the appropriateness of the content to the local context, two experts in diabetes (a family medicine specialist and an endocrinologist) and an expert in questionnaire translation reviewed and discussed the items of the instrument. The revised single forward translation was then proofread by a Malay language expert to eliminate grammatical error.

Ten adults with T2DM were recruited for cognitive debriefing of the prefinal version of the DHLS-M. They were requested to answer the questionnaire and asked if they understood the words or terms and sentences in the questionnaire. The comments received during the cognitive debriefing were reviewed, and changes were made. The changes were verified by four adults with T2DM. The final version of the DHLS-M was generated.

## Data collection

Ethical approval for this study was obtained from the Human Research Ethics Committee USM prior to data collection ((USM/JEPeM/KK/23080652). This study was conducted following the Declaration of Helsinki. Data collection was performed using a self-administered questionnaire. After the potential participants obtained the queue number for their outpatient visit, they were approached and invited to participate in this study. Eligibility criteria were checked and a copy of the participant information sheet, informed consent form (ICF) and questionnaire was given to them. They were briefed about the study and informed to fill up the ICF and the questionnaire if they agreed to participate. The return of the signed ICF and the questionnaire was considered complete participation. Assistance (read the questionnaire aloud) was provided for the participants with long-sightedness and did not bring their glasses. Clarification was given if the participants have any query. The estimated time to complete the questionnaire was 20 to 30 min. Lastly, the participants' HbA1c level was obtained from their laboratory records.

## Statistical analysis

The descriptive statistics was performed using IBM SPSS Statistics (version 27). The characteristics of the participants were described as frequency (percentage) for categorical variable and median (IQR) for non-normally distributed numerical variable. The univariate normality was determined visually using a histogram.

The CFA was conducted using Mplus 8.0 to determine the construct validity and reliability of the DHLS-M. The multivariate normality assumption was examined using Mardia's Test for Multivariate Normality (Skewness and Kurtosis). The results ($p < 0.05$) showed that the data did not meet the multivariate normality assumption. As the data was non-normally distributed and ordinal, the robust weighted least squares (WLSMV) estimator was used for the analysis. *Hair et al. (2014)* suggested using three to four fit indices with at least one absolute fit index and one incremental fit index to assess the model fit. The model fit was assessed based on the following fit indices and cut-off values: $\chi^2/df$ of less than 5 (*Wheaton et al., 1977*), RMSEA of less than 0.08, CFI and Tucker–Lewis index (TLI) of more than or equals to 0.95 (*Hair et al., 2014*). $\chi^2/df$ instead of $\chi^2$ was used to

reduce the impact of sample size (*Hooper, Coughlan & Mullen, 2008*). The factor loading of each item was assessed. Item with factor loading of less than 0.4 and more than 1.0 could be considered for deletion with theoretical support (*DeVon et al., 2007*; *Hair et al., 2014*). For models with three subscales, the factor correlation of each pair of subscales was assessed. A value of more than 0.85 indicated the two subscales were correlated and suggested to collapse the subscales with theoretical support (*Brown, 2015*). The model was re-specified based on the model fit, factor loading and factor correlation as well as discussion among the researchers. The composite reliability based on Raykov's rho was determined for each subscale in the final model. A value of 0.7 or higher indicated good reliability (*Hair et al., 2014*).

Spearman's correlation was carried out using IBM SPSS Statistics (version 27) to determine the correlation between the score of the Malay version of the MDKT, the subscales and the health literacy index of the Malay version of the HLS-SF12 and the subscales of the DHLS-M. A significant correlation established the convergent validity between the Malay version of the MDKT, the Malay version of the HLS-SF12 and the DHLS-M. The correlation coefficient was interpreted following the recommendation by *Cohen (1988)*: weak correlation (0.10–0.29), moderate correlation (0.30–0.49) and strong correlation (0.50–1.00).

## RESULTS

### Questionnaire translation and adaptation

During the questionnaire translation and adaptation process, five changes were made to the English version of the DHLS. The statement in the instruction was changed from 'there is no correct answer for any of the items' to 'there is no right or wrong answer for all the items'. The response scale was changed from 'not really, slightly, moderately, quite a lot, very much' to 'strongly disagree, disagree, partly agree, agree, strongly agree'. The term 'healthcare provider' was changed to 'healthcare practitioner (*e.g.*, doctor, nurse, pharmacist, *etc.*)'. Examples were added to item A3, changed from 'I comprehend the information I sought on diabetes' to 'I comprehend the information I sought on diabetes (*e.g.*, from internet, newspaper, magazine, *etc.*)'. Item A6 which stated as 'I can print out my prescription from an automated prescription machine at the hospital' was adapted to 'I know how to obtain my prescription from the hospital' to suit the local context.

### Characteristics of the participants

A total of 250 participants with complete data on DHLS-M were included in the analysis. The characteristics of the participants were summarized in Table 1. The median age of the participants was 63.0 years old (IQR = 12.3). One hundred and twenty-eight (128) participants (51.2%) were male and 122 participants (48.8%) were female. Most of the participants were Malay (92.4%). For education level, 52.0% of the participants received secondary education, followed by tertiary education (41.6%), primary education (4.8%) and no formal education (1.6%).

**Table 1  Characteristics of the participants ($n = 250$).**

| Variables | Frequency (%) | Median (IQR) |
|---|---|---|
| **Age, years** | | 63.0 (12.3)[*] |
| **Disease duration, years** ($n = 244$) | | 12.0 (12.8)[**] |
| **Gender** | | |
| Male | 128 (51.2) | |
| Female | 122 (48.8) | |
| **Race** | | |
| Malay | 231 (92.4) | |
| Chinese | 18 (7.2) | |
| Indian | 1 (0.4) | |
| **Marital status** | | |
| Married | 215 (86.0) | |
| Divorcee/widow(er) | 28 (11.2) | |
| Single | 7 (2.8) | |
| **Education level** | | |
| No formal education | 4 (1.6) | |
| Primary education | 12 (4.8) | |
| Secondary education | 130 (52.0) | |
| Tertiary education | 104 (41.6) | |
| **Occupation** | | |
| Working | 69 (27.6) | |
| Pensioner | 124 (49.6) | |
| Not working | 57 (22.8) | |
| **Monthly income, RM** | | |
| Less than 3,000 | 163 (65.2) | |
| 3,001–5,000 | 39 (15.6) | |
| 5,001–7,000 | 31 (12.4) | |
| More than 7,000 | 17 (6.8) | |
| **Treatment regimen** | | |
| Oral hypoglycemic agent | 96 (38.4) | |
| Insulin | 15 (6.0) | |
| Oral hypoglycemic agent + insulin | 138 (55.2) | |
| Oral hypoglycemic agent + semaglutide | 1 (0.4) | |
| **HbA1c level** | | |
| Controlled, HbA1c < 7.0% | 47 (18.8) | |
| Uncontrolled, HbA1c ≥ 7.0% | 203 (81.2) | |

Notes.
[*]Skewed to the left.
[**]Skewed to the right.

## CFA of DHLS-M

The initial model, which consists of three subscales and 14 items showed that the fit indices were not within the acceptable threshold value except for CFI and TLI. The results of fit indices were displayed in Table 2 (see model 'Initial'). The factor loading of the items were inspected and shown in Table 3 (see model 'Initial'). The factor loading of item

**Table 2   CFA's fit indices for measurement model of DHLS-M.**

| Model | $\chi^2/df$ | RMSEA (90% CI) | CFI | TLI |
|---|---|---|---|---|
| Initial | 5.046 | 0.127 (0.115, 0.140) | 0.976 | 0.970 |
| Revised[a] | 3.858 | 0.107 (0.093, 0.121) | 0.981 | 0.976 |

**Notes.**
$\chi^2/df$, Relative Chi-square; RMSEA, Root Mean Square Error of Approximation; CFI, Comparative Fit Index; TLI, Tucker-Lewis Index; CI, Confidence Interval.
[a]Model with B1 removed.

B1 was 1.008, which was considered as a problematic item. After discussion among the researchers, item B1 ('I can calculate the next time to take diabetes medication') was removed as calculation of next dosing time was not required when simple instruction on the medication dosing time was given by the healthcare provider. The model was re-specified. The factor loading of all the items in the revised model were statistically significant and in the range of 0.751 to 0.979. The fit indices of the revised model were improved ($\chi^2/df$ = 3.858, RMSEA (90% CI) = 0.107 (0.093, 0.121), CFI = 0.981, TLI = 0.976). The $\chi^2/df$, CFI, TLI showed acceptable model fit while RMSEA was higher than the acceptable threshold value. The factor correlation of each pair of subscales in the revised model were examined and tabulated in Table 4. The correlation of each pair of subscales were less than 0.85 except the correlation between informational health literacy and communicative health literacy subscale ($r$ = 0.903). After discussion among the researchers, it was decided not to combine the two subscales but to maintain the original structure of the 3-subscale model. The retention of the two highly correlated subscales most likely caused the RMSEA to be higher than the acceptable threshold value. The composite reliability was computed for each subscale in the revised model and presented in Table 4. The composite reliability of the three subscales were 0.962, 0.839 and 0.828, respectively. Good reliability was demonstrated. The revised model with 13 items and three subscales was accepted as the final model.

### Correlation between DHLS-M and Malay version of MDKT

A total of 247 participants with complete data on DHLS-M and the Malay version of the MDKT were included in the analysis. The subscales of the DHLS-M were weakly to moderately correlated with the Malay version of the MDKT ($r$ = 0.26 to 0.31; $p < 0.001$). The results were presented in Table 5.

### Correlation between DHLS-M and Malay version of HLS-SF12

A total of 243 participants with complete data on DHLS-M and the Malay version of the HLS-SF12 were included in the analysis. The subscales of the DHLS-M were moderately to strongly correlated with the subscales and the health literacy index of the Malay version of the HLS-SF12 ($r$ = 0.43 to 0.66; $p < 0.001$). The results were presented in Table 5.

## DISCUSSION

This study has translated, adapted and validated a diabetes-specific health literacy instrument, DHLS-M. The English version of the DHLS was translated into Malay and

**Table 3   CFA's standardized factor loading of DHLS-M.**

| Subscale | Model | Initial | Revised |
|---|---|---|---|
| | Item | Standardized factor loading | |
| Informational health literacy | A1 | 0.878 | 0.897 |
| | A2 | 0.937 | 0.979 |
| | A3 | 0.832 | 0.844 |
| | A4 | 0.895 | 0.889 |
| | A5 | 0.817 | 0.837 |
| | A6 | 0.898 | 0.865 |
| | A7 | 0.901 | 0.877 |
| Numerate health literacy | B1 | 1.008 | – |
| | B2 | 0.715 | 0.794 |
| | B3 | 0.670 | 0.760 |
| | B4 | 0.747 | 0.836 |
| Communicative health literacy | C1 | 0.817 | 0.808 |
| | C2 | 0.790 | 0.795 |
| | C3 | 0.748 | 0.751 |

Notes.

All factor loadings were statistically significant ($p < 0.001$), (-) = removed item.

**Table 4   Factor correlation and composite reliability based on the measurement model of DHLS-M.**

| Subscale | Composite reliability | Factor correlation | | |
|---|---|---|---|---|
| | | 1 | 2 | 3 |
| 1. Informational health literacy | 0.962 | 1 | 0.836[*] | 0.903[*] |
| 2. Numerate health literacy | 0.839 | | 1 | 0.839[*] |
| 3. Communicative health literacy | 0.828 | | | 1 |

Notes.

[*]All factor correlations were statistically significant ($p < 0.001$).

**Table 5   Correlation between Malay version of MDKT, Malay version of HLS-SF12 and DHLS-M, to describe the convergent validity of the questionnaires' scores.**

| Variables | DHLS-M (Correlation coefficient, $r$) | | |
|---|---|---|---|
| | Informational health literacy | Numerate health literacy | Communicative health literacy |
| **MDKT** ($n = 247$) | 0.30 | 0.31 | 0.26 |
| **HLS-SF12** ($n = 243$) | | | |
| Health care | 0.60 | 0.61 | 0.45 |
| Health prevention | 0.58 | 0.62 | 0.46 |
| Health promotion | 0.55 | 0.57 | 0.43 |
| Health literacy index | 0.63 | 0.66 | 0.49 |

Notes.

All correlations were statistically significant ($p < 0.001$).

adapted to local context based on the translation and adaptation process in *Wild et al. (2005)* with some modification to ensure the equivalency between the English and the Malay versions. Five changes were made during the translation and adaptation process of the English version of the DHLS to Malay due to the unavailability of the Malay term with exact same meaning in English, to increase the clarify of the statement or sentence, to increase the understanding of the study participants and to adapt to the local context. *Moshki et al. (2022)* also reported that four questions were modified when the English version of the DHLS was translated into Persian.

CFA was performed to confirm the construct validity and reliability of the DHLS-M. The model fit of the initial model was insufficient. A problematic item, item B1 was identified and removed. The model fit of the revised model was improved with three fit indices ($\chi^2/df$, CFI, TLI) within the acceptable threshold value, and accepted as the final model. The factor loading of all the items were more than 0.4 and statistically significant. The correlation between each pair of subscales were less than 0.85 except the correlation between informational health literacy and communicative health literacy subscale. The composite reliability of the three subscales were more than 0.7, showed good reliability.

The problematic item B1 ('I can calculate the next time to take diabetes medication') with factor loading more than one was removed after discussion among the researchers. This item may not be relevant to adults with T2DM as diabetes is a chronic disease and the patients are on long-term diabetes medications. The patients have been educated on the right time to take their medications. *Lee et al. (2020)* also mentioned that the item related to the calculation of the next dosing time of medication or the time to take the next dose was no longer appropriate to be included in a health numeracy instrument in South Korea as the dosing time of the medication has been printed clearly on the label.

The informational health literacy subscale was highly correlated with the communicative health literacy subscale. The items in informational health literacy subscale were related to the ability to read, understand and judge diabetes-related information as well as to use the healthcare services. On the other hand, the items in communicative health literacy subscale were related to the ability to ask and share diabetes-related information with healthcare providers and people around. As the two subscales measuring two different dimensions of health literacy, both subscales were retained. The Persian version of the DHLS also found a high correlation between two subscales (informational health literacy and numerate health literacy) and both subscales were retained (*Moshki et al., 2022*).

For the final model, three fit indices ($\chi^2/df$, CFI, TFI) were within the acceptable threshold. The RMSEA was higher than the acceptable threshold. RMSEA is affected by the complexity of the model (*Brown, 2015*). The high RMSEA was likely due to the highly correlated subscales (informational health literacy and communicative health literacy). As the original Korean version and the Persian version of the DHLS were three-subscale model (*Lee et al., 2018*; *Moshki et al., 2022*), further re-specification of the model was not performed.

Good reliability was demonstrated with composite reliability of more than 0.7 for the three subscales. However, the composite reliability of the informational health literacy subscale was more than 0.95. Composite reliability of more than 0.95 indicates all the

items are measuring the same thing or the items are redundant (*Hair et al., 2017*). The word 'understand' was used in three items and the word 'comprehend' was used in an item, but each item covered different aspects of information on diabetes. Item A1 refers to educational materials, item A2 refers to diabetes treatment or examination, item A4 refers to diabetes management and item A3 refers to information from various sources sought by the participants. Therefore, no item was deleted after discussion among the researchers. The final model consisted of three subscales with seven, three and three items, respectively.

Spearman's correlation was conducted to determine the relationship between the Malay version of the MDKT, the Malay version of the HLS-SF12 and the DHLS-M. The subscales of the DHLS-M were weakly to moderately correlated with diabetes knowledge which was measured with the Malay version of the MDKT. The same finding has been shown in the original Korean version of the DHLS where DHLS was moderately correlated with diabetes knowledge (*Lee et al., 2018*). A meta-analysis conducted by *Marciano, Camerini & Schulz (2019)* reported that performance-based health literacy instrument was more strongly correlated with diabetes knowledge compared to self-reported health literacy instrument. This further explained the weakly to moderately relationship between the DHLS-M, which is a self-reported diabetes-specific health literacy instrument and diabetes knowledge.

In addition, the subscales of the DHLS-M were moderately to strongly correlated with the subscales and health literacy index of the Malay version of the HLS-SF12. The HLS-SF12 is used to measure general health literacy among the adults (*Duong et al., 2019*). The correlation between diabetes health literacy and general health literacy was also reported in the original Korean version of DHLS (*Lee et al., 2018*).

There were several limitations in this study. The data were collected from a single hospital. More than 90% of the participants were Malay. The non-Malay participants in this study were less than 10%. Thus, the results cannot be generalised to the whole diabetes population in Malaysia. Therefore, it is recommended to cross-validate DHLS-M in a study with a larger sample size, higher percentage of non-Malay participants in other states or several states in Malaysia that are more reflective of the diabetes population in Malaysia. The problematic item in this study should be examined again in other diabetes populations. This study was conducted using a self-administered questionnaire which consisted of two self-reported instruments and a performance-based instrument. For self-reported instruments, participants tend to provide social desired or positive responses which result in overestimation of health literacy. However, self-reported instruments are still widely accepted for research purposes or in healthcare settings. Furthermore, some of the participants required assistance to read the questionnaire aloud and sought clarification during the administration of the questionnaire. This may introduce bias to the study. A bigger font size was used in printing the questionnaire to reduce the number of participants who needed help during the administration of the questionnaire. Consistent clarification was given to the participants and generally the participants sought clarification on the sociodemographic section.

## CONCLUSIONS

This study successfully translated and adapted the DHLS to Malay, followed by validation of the DHLS-M. The translation and adaptation process ensured that the DHLS-M was equivalent with the English version of the DHLS and relevant to the adults with diabetes in Malaysia. The final model of DHLS-M showed an acceptable model fit, valid and reliable although the informational health literacy subscale was highly correlated with the communicative health literacy subscale, and the items in the informational health literacy subscale may be redundant. The final model of DHLS-M consisted of three subscales with 13 items. The validity of DHLS-M was further strengthened by the convergent validity between the Malay version of the MDKT, the Malay version of the HLS-SF12 and the DHLS-M. DHLS-M can be used to measure diabetes health literacy among adults with diabetes in Malaysia. DHLS-M can also be used in study to identify factors associated with diabetes health literacy so that action can be taken by the healthcare providers to improve the health literacy among the adults with diabetes.

## ACKNOWLEDGEMENTS

We wish to thank all the participants in this study.

### Funding

The authors received no funding for this work.

### Competing Interests

The authors declare there are no competing interests.

### Author Contributions

- Pei Ling Ng conceived and designed the experiments, performed the experiments, analyzed the data, prepared figures and/or tables, authored or reviewed drafts of the article, and approved the final draft.
- Wan Nor Arifin conceived and designed the experiments, authored or reviewed drafts of the article, and approved the final draft.
- Wan Mohd Izani Wan Mohamed conceived and designed the experiments, authored or reviewed drafts of the article, and approved the final draft.
- Rosnani Zakaria conceived and designed the experiments, authored or reviewed drafts of the article, and approved the final draft.
- Yee Cheng Kueh conceived and designed the experiments, analyzed the data, authored or reviewed drafts of the article, and approved the final draft.

### Human Ethics

The following information was supplied relating to ethical approvals (i.e., approving body and any reference numbers):

Universiti Sains Malaysia Human Ethics Committee.

## Data Availability

The data is available in the Supplemental File.

## Supplemental Information

Supplemental information for this article can be found online at http://dx.doi.org/10.7717/peerj.19660#supplemental-information.

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
