# Peer review of "Validation of the Malay version of the Diabetes Health Literacy Scale among Malaysian adults with type 2 diabetes mellitus"

_PeerJ, doi:10.7717/peerj.19660_

## Round 0.1 · original submission · Minor Revisions

Dear Dr. Kueh,
Thank you for your submission to PeerJ.
It is my opinion as the Academic Editor for your article - Validation of the Malay version of the diabetes health literacy scale among Malaysian adult with type 2 diabetes mellitus - that it requires a few Minor Revisions.
Warm regards

Reviewer 1 ·

Basic reporting

The introduction provides sufficient context, but it would be helpful to discuss why the DHLS is particularly important for the Malaysian population.

The discussion of health literacy’s impact on diabetes self-management is well-referenced. However, you may consider strengthening the argument with additional literature on the role of culture-specific adaptations in health literacy instruments.

The conclusion in the abstract could be more informative for better understanding.

Experimental design

The manuscript by Ng titled "Validation of the Malay version of the diabetes health literacy scale among Malaysian adults with type 2 diabetes mellitus" aims to validate the Malay version of the Diabetes Health Literacy Scale (DHLS-M), addressing a clear research gap.

For the method, the exclusion criteria could be further justified. Why were patients with gestational diabetes excluded? Would their inclusion significantly affect results?

Validity of the findings

In lines 238-255, for the CFA analysis, the fit indices are reported, but a justification for why RMSEA remained above the threshold should be included. In addition, for Table 2, explaining the rationale for removing item B1 earlier in the text would improve clarity.

Reviewer 2 ·

Basic reporting

The references provided throughout the manuscript are sufficient, however, most of them are older than the year 2020. I suggest that the authors review and update, where possible, for the last 5 years.

The manuscript is well structured and presents sufficient data, but I suggest changing the position of the first two paragraphs of the introduction, starting with the definition of diabetes and continuing with epidemiological data.

The tables are clear and objective.

The results are relevant to support the hypothesis presented.

Experimental design

The authors present in the abstract that the objective of the work was to verify the validity and reliability of the instrument, however, in line 80 of the introduction, the authors inform that the objective of the study was to translate and adapt the DHLS to the Malay language, in addition to the validity and reliability. Was the objective of the work to translate and validate or just validate? I suggest standardizing the objectives throughout the text. Considering that the authors report that the instrument was translated and adapted, in addition to its validity and reliability, wouldn't this be a methodological study? I believe that the cross-sectional study design would not cover the translation and adaptation stage of the instrument.

Was any instrument used to measure whether the participants were able to read, understand and answer the questionnaire? Or were they simply asked if they were able? I suggest reviewing the exclusion criteria, since patients with DM1 and gestational diabetes were not part of the study population, and therefore cannot be considered as exclusion criteria, since they were not even included in the sample.

In line 148, the authors describe the process of translating and adapting the instrument from English to Malay. As previously mentioned, it is necessary to make it clear in the objectives that the translation process was also carried out.

In line 270 of the discussion, the authors report that the study validated an instrument, however, they do not mention the translation. Rephrase the sentence, making it clear that the translation and adaptation were also carried out during the development of the study.

Validity of the findings

The conclusions are well formulated and linked to the original question. However, as previously mentioned, the authors inform throughout the text that the translation and adaptation stage was also carried out, and therefore, they must also present conclusions for this purpose.

Additional comments

I congratulate the authors for the excellent work developed. I believe that the data are relevant and the instrument can be used in the development of future research in the area of ​​diabetes in Malaysia. However, some adjustments throughout the manuscript need to be made, in order to make it clearer, so that it can be replicated by other researchers in the future.

---

## Round 0.2 · accepted · Accept

Dear Authors

Your revision has been reviewed by the experts in the field, and we recognize that the revision is sufficiently justified and has been improved. Your submission is now endorsed by two experts for acceptance of publication in PeerJ.

Thank you for submitting your article to PeerJ. I would like to express my gratitude for your contributions and efforts to the scientific community. I look forward to receiving your research and review articles in the future.

Best Regards

Yung-Sheng Chen, Ph.D.
Academic Editor

Reviewer 1 ·

Basic reporting

No further changes are required, the authors have appropriately revised the manuscript as suggested.

Experimental design

Experimental design is sufficiently justified and clearly described.

Validity of the findings

The authors have addressed all prior concerns.